# Real World Offline Reinforcement Learning with Realistic Data Source

**Gaoyue Zhou**[1*]     **Liyiming Ke**[2*]     **Siddhartha Srinivasa**[2]     **Abhinav Gupta**[1]

**Aravind Rajeswaran**[3]                    **Vikash Kumar**[3]

[1]Carnegie Mellon University        [2]University of Washington        [3]Meta AI

## Abstract

Offline reinforcement learning (ORL) holds great promise for robot learning due to its ability to learn from arbitrary pre-generated experience. However, current ORL benchmarks are almost entirely in simulation and utilize contrived datasets like replay buffers of online RL agents or sub-optimal trajectories, and thus hold limited relevance for real-world robotics. In this work (Real-ORL), we posit that data collected from safe operations of closely related tasks are more practical data sources for real-world robot learning. Under these settings, we perform an extensive (6500+ trajectories collected over 800+ robot hours and 270+ human labor hour) empirical study evaluating generalization and transfer capabilities of representative ORL methods on four real-world tabletop manipulation tasks. Our study finds that ORL and imitation learning prefer different action spaces, and that ORL algorithms can generalize from leveraging offline heterogeneous data sources and outperform imitation learning. We release our dataset and implementations at URL: `https://sites.google.com/view/real-orl`

## 1   Introduction

Despite rapid advances, the applicability of Deep Reinforcement Learning (DRL) algorithms [1–8] to real-world robotics tasks is limited due to sample inefficiency and safety considerations. The emerging field of offline reinforcement learning (ORL) [9, 10] has the potential to overcome these challenges, by learning only from logged or pre-generated offline datasets, thereby circumventing safety and exploration challenges. This makes ORL well suited for applications with large datasets (e.g. recommendation systems) or those where online interactions are scarce and expensive (e.g. robotics). However, comprehensive benchmarking and empirical evaluation of ORL algorithms is significantly lagging behind the burst of algorithmic progress [11–21]. Widely used ORL benchmarks [22, 23] are entirely in simulation and use contrived data collection protocols that do not capture fundamental considerations of physical robots. Our work aims to bridge this gap by outlining practical offline dataset collection protocols that are representative of real-world robot settings. Our work also performs a comprehensive empirical study spanning **6500+ robot trajectories** and **270+ human labor hours**, to benchmark and analyze three representative ORL algorithms thoroughly. We will release all the datasets, code, and hardware hooks from this paper.

In principle, ORL can consume and train policies from arbitrary datasets. This has prompted the development of simulated ORL benchmarks [22–24] that utilize data sources like expert policies trained with *online* RL, exploratory policies, or even replay buffers of *online* RL agents. However, simulated dataset may fail to capture the challenges in real world: hardware noises coupled with

---

*Equal contribution.

Offline Reinforcement Learning Workshop at Neural Information Processing Systems, 2022

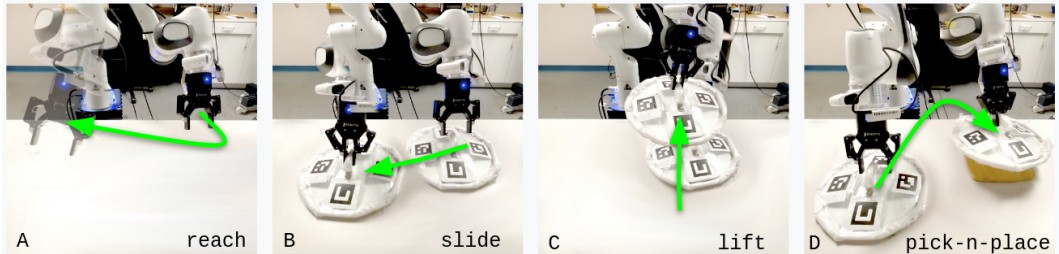

Figure 1: Canonical tasks for tabletop manipulation.

varying reset conditions lead to covariate shift and violate the i.i.d. assumption about state distributions between train and test time. Further, such datasets are not feasible on physical robots and defeat the core motivation of ORL in robotics – to avoid the use of *online* RL due to poor sample efficiency and safety! Recent works [24, 25] suggest that dataset composition and distribution dramatically affect the relative performance of algorithms. In this backdrop, we consider the pertinent question: *What is a practical instantiation of the ORL setting for physical robots, and can existing ORL algorithms learn successful policies in such a setting?*

In this work, we envision *practical* scenarios to apply ORL for real-world robotics. Towards this end, our first insight is that real-world offline datasets are likely to come from well-behaved policies that abide by safety and monetary constraints, in sharp contrast to simulator data collected from exploratory or partially trained policies, as used in simulated benchmarks [22–24]. Such trajectories can be collected by user demonstrations or through hand-scripted policies that are partially successful but safe. It is more realistic to collect large volumes of data for real robots using multiple successful policies designed under expert supervision for specific tasks than using policies that are unsuccessful or without safety guarantees. Secondly, the goal of any learning (including ORL) is broad generalization and transfer. It is therefore critical to study whether a learning algorithm can leverage task-agnostic datasets, or datasets intended for a source task, to make progress on a new target task. In this work, we collect offline datasets consistent with these principles and evaluate representative ORL algorithms on a set of canonical table-top tasks as illustrated in Figure 1.

Evaluation studies on physical robots are sparse in the field due to time and resource constraints, but they are vital to furturing our understanding. Our real robot results corroborate and validate intuitions from simulated benchmarks [26] but also enable novel discoveries. We find that (1) even for scenarios with sufficiently high-quality data, some ORL algorithms could outperform behavior cloning (BC) [27] on specific tasks, (2) for scenarios that require generalization or transfer to new tasks with low data support, ORL agents generally outperform BC. (3) in cases with overlapping data support, ORL algorithms can leverage additional heterogeneous task-agnostic data to improve their own performance, and in some cases even surpass the best in-domain agent.

Our empirical evaluation is unique as it focuses on ORL algorithms ability to leverage more realistic, multi-task data sources, spans over several tasks that are algorithm-agnostic, trains various ORL algorithms on the same settings and evaluates them directly in the real world. In summary, we believe our work establishes the effectiveness of offline RL algorithms in leveraging out of domain high-quality heterogeneous data for generalization and transfer in robot-learning, which is representative of real world applications.

## 2 Preliminaries and Related Work

**Offline RL.** We consider the ORL framework, which models the environment as a Markov Decision Process (MDP): $M = \langle S, A, R, T, \rho_0, H \rangle$ where $S \subseteq \mathbb{R}^n$ is the state space, $A \subseteq \mathbb{R}^m$ is the action space, $R : S \times A \to \mathbb{R}$ is the reward function, $T : S \times A \times S \to \mathbb{R}_+$ is the (stochastic) transition dynamics, $\rho_0 : S \to \mathbb{R}_+$ is the initial state distribution, and $H$ is the maximum trajectory horizon. In the ORL setting, we assume access to the reward function $R$ and a pre-generated dataset of the form: $\mathbb{D} = \{\tau_1, \tau_2, \ldots \tau_N\}$, where each $\tau_i = (s_0, a_0, s_1, a_1, \ldots s_H)$ is a trajectory collected using a behavioral policy or a mix of policies $\pi_b : S \times A \to \mathbb{R}_+$.

The goal in ORL is to use the offline dataset $\mathbb{D}$ to learn a near-optimal policy,

$$\pi^* := \arg\max_{\pi} \; \mathbb{E}_{M,\pi} \left[ \sum_{t=0}^{H} r(s_t, a_t) \right].$$

In the general case, the optimal policy $\pi^*$ may not be learnable using $\mathbb{D}$ due to a lack of sufficient exploration in the dataset. In this case, we would seek the best policy learnable from the dataset, or, at the very least, a policy improves upon behavioral policy.

**Offline RL Algorithms.** Recent years have seen tremendous interests in offline RL and the development of new ORL algorithms. Most of these algorithms incorporate some form of regularization or conservatism. This can take many forms, such as regularized policy gradients or actor critic algorithms [14, 15, 19, 28–30], approximate dynamic programming [11, 13, 17, 18], and model-based RL [12, 31–33]. We select a representative ORL algorithms from each category: AWAC [19], IQL [18] and MOREL [12]. In this work, we do not propose new algorithms for offline RL; rather we study a spectrum of representative ORL algorithms and evaluate their assumptions and effectiveness on a physical robot under realistic usage scenarios.

**Offline RL Benchmarks and Evaluation.** In conjunction with algorithmic advances, offline RL benchmarks have also been proposed. However, they are predominantly captured with simulation [22, 23, 34] using datasets with idealistic coverage, i.i.d. samples, and synchronous execution. Most of these assumptions are invalid in real world which is stochastic and has operational delays. Prior works investigating offline RL for these settings on physical robots are limited. For instance, Kostrikov et al. [18] did not provide real robot evaluation for IQL, which we conduct in this work; Chebotar et al. [35], Kalashnikov et al. [36] evaluate performance on a specialized Arm-Farm; Rafailov et al. [37] evaluate on a single drawer closing task; Singh et al. [17], Kumar et al. [38] evaluate only one algorithm (COG, CQL, respectively). Mandlekar et al. [39] evaluate BCQ and CQL alongside BC on three real robotics tasks, but their evaluations consider only in-domain setting: that the agents were trained only on the specific task data, without giving them access to a pre-generated, offline dataset. Thus, it is unclear whether insights from simulated benchmarks or limited hardware evaluation can generalize broadly. Our work aims to bridge this gap by empirically studying representative offline RL algorithms on a suite of real-world robot learning tasks with an emphasize on transfer learning and out-domain generalization. See Section 3 for detailed discussion.

**Imitation Learning (IL).** IL [40] is an alternate approach to training control policies for robotics. Unlike RL, which learns policies by optimizing rewards (or costs), IL (and inverse RL [41–43]) learns by mimicking expert demonstrations and typically requires no reward function. IL has been studied in both the offline setting [44, 45], where the agent learns from a fixed set of expert demonstrations, and the online setting [46, 47], where the agent can perform additional environment interactions. A combination of RL and IL has also been explored in prior work [48, 49]. Our offline dataset consists of trajectories from a heuristic hand-scripted policy collected under expert supervision, which represents a dataset of reasonably *high quality*. As a result, we consider offline IL and, behavior cloning in particular, as a baseline algorithm in our empirical evaluation.

## 3 Experiment Scope and Setup

To investigate the effectiveness of ORL algorithms on real-world robot learning tasks, we adhere to a few guiding principles: (1) we make design choices representing the wider community to the extent possible, (2) we strive to be fair to all baselines by providing them their best chance and work in consultation with their authors; and (3) we prioritize reproducibility and data sharing. We will open-source our data, camera images along with our training and evaluation codebase.

**Hardware Setup.** Hardware plays a seminal role in robotic capability. For reproducibility and extensibility, we selected a hardware platform that is well-established, non-custom, and commonly used in the field. After an exhaustive literature survey [50–55], we converged on a table-top manipulation setup, shown in Figure 2. It consists of a table-mounted Franka panda arm that uses a RobotiQ parallel gripper as its end effector, which is accompanied by two Intel 435 RGBD cameras. Our robot has 8 DOF, uses factory-supplied default controller gains, accepts position commands at 15

Hz, and runs a low-level joint position controller at 1000 Hz. To perceive the object to interact with, we exact the position of the AprilTags attached to the object from RGB images. Our robot states consist of joint positions, joint velocities, and positions of the object to interact with (if applicable). Our policies compute actions (desired joint pose) using robot proprioception, tracked object locations, and desired goal location.

**Canonical Tasks**   We consider four classic manipulation tasks common in literature: `reach`, `slide`, `lift`, and `pick-n-place` (PnP) (see Figure 1). `reach` requires the robot to move from a randomly sampled configuration in the workspace to another configuration. The other three tasks involve a heavy glass lid with a handle, which is initialized randomly on the table. `slide` requires the robot to hold and move the lid along the table to a specified goal location. `lift` requires the robot to grasp and lift the lid 10 cm off the table. PnP requires the robot to grasp, lift, move and place the lid at a designated goal position i.e. the chopping board. The four tasks constitute a representative range of common tabletop

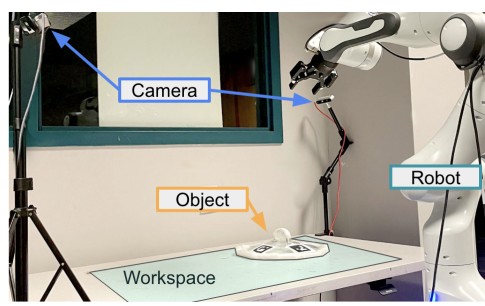

Figure 2: Our setup consists of a commonly used Franka arm, a RobotiQ parallel gripper, and two Intel Realsense 435 cameras.

manipulation challenges: `reach` focuses on free movements while the other three tasks involve intermittent interaction dynamics between the table, lid, and the parallel grippers. We model each canonical task as a MDP with an unique reward function. Details on our tasks are in Appendix. 7.1.

**Data Collection.**   We use a hand-designed, scripted policy developed under expert supervision to collect (dominantly) successful trajectories for all our canonical tasks. To highlight ORL  algorithms ability to overcome suboptimal dataset, previous works [22, 34, 39] have crippled expert policies with noise, use half-trained RL policies or collect human demonstrations with varying qualities to highlight the performance gain over compromised datasets. We posit that such data sources are not representative of robotics domains, where noisy or random behaviors are unsafe and detrimental to hardware's stability. Instead of infusing noise or failure data points to serve as negative examples, we believe that mixing data collected from various tasks offers a more realistic setting in which to apply ORL on real robots for three reasons: (1) collecting such "random/roaming/explorative" data on a real robot autonomously would require comprehensive safety constraints, expert supervision and oversight, (2) engaging experts to record such random data in large quantities makes less sense than utilizing them to collecting meaningful trajectories on a real task, and (3) designing task-specific strategies and stress testing ORL's ability against such a strong dataset is more viable than using a compromised dataset. We collected offline dataset using heuristic strategies designed with reasonable efforts and, to avoid biases favoring task/algorithm, frozen the dataset ahead of time.

**Dataset.**   In total, we collected a dataset of around 3000 trajectories and the characteristics of our dataset is shown in Table. 1. Our offline dataset is represented as a series of transition tuples $\{(s, a, s')_{task}\}$. States consist of joint positions, joint velocities, and positions of the object to interact with (if applicable). Actions contain target joint positions. To perceive the object to interact with, we obtain the position of tracked AprilTags attached to the object from the RGB images of the two cameras. More details are available in Appendix 7.2.

| Task | # Traj | # Samples | Avg Score | Max Score | Human Score | Theoretical Best Score |
|------|--------|-----------|-----------|-----------|-------------|------------------------|
| reach | 1000 | 99752 | 0.960 | 0.99 | 0.963 | 1 |
| slide | 731 | 244422 | 0.819 | 0.93 | 0.834 | 1 |
| lift | 609 | 178515 | 0.948 | 1 | 1 | 1 |
| PnP | 616 | 327478 | 0.875 | 1.09 | 0.924 | 1.15 |

Table 1: Characteristics of collected data. *# Traj* denotes the total number of trajectories, *# Samples* denotes the total number of state-action-reward pairs. Each trajectory's score is the maximum reward in the trajectory. *Avg Score* shows the average scores per trajectories, *Max Score* shows the maximum reward achieved by trajectories in our dataset, *Human Score* shows the max reward achieved by a human teleoperator and *Theoretical Best Score* denotes the theoretical maximum possible reward determined by our reward function.

# 4 Experiment Design

Our experiments aim to answer the following questions. **(1)** Are ORL algorithms sensitive to, or show a preference for, any specific state and action space parameterization? **(2)** How do they perform against the standard methods for in-domain tasks? **(3)** How do common methods perform in out-of-domain tasks requiring (a) generalization, and (b) re-targeting? To ensure fair evaluation, we now outline our choice of candidate algorithms and performance metrics.

**Algorithms**  For all evaluations, we compare four algorithms: Behavior Cloning (BC) [27], Model-based Offline REinforcement Learning (MOREL) [12], Advantage-Weighted Actor Critic (AWAC) [19] and Implicit Q-Learning (IQL) [18]. BC is a model-free IL algorithm that remains a strong baseline for real robot experiments due to its simplicity and practicality. AWAC and IQL both train an off-policy value function and then derive a policy to maximize the expected reward. AWAC uses KL divergence minimization to constrain the resulting policy to be close to the given policy distribution. In contrast, IQL leverages expectile regression to avoid querying the value function for any out-of-distribution query. MOREL is distinct since it is a model-based approach: it recovers a dynamics model from offline data that allows it directly apply policy gradient RL algorithms. We use implementations of BC and MOREL from the MOREL author implementation. For the later, we add a weighted behavior cloning loss to its policy training step to serve as a regularizer, inspired by [30]. We use AWAC and IQL implemented in the open sourced d3rlpy library [56].

**Training.**  Since neural network agents are empirically sensitive to parameters and seeds, we (1) used the same fixed random seed (123) for all our experiments with additional seed sweeping to strengthen the reproducibility of our results and (2) conducted equal amount of efforts for hyperparameter tuning efforts for all algorithms. Unlike traditional supervised learning, we cannot simply select the agents with the best validation loss for tuning the hyperparameters, because we cannot know the performance of an agent unless testing it on a real robot [39]. We thus keep our tuning simple and fair: starting with the default parameters and training 5 agents in 3 rounds, trying to make the agent converge. We observe that certain agents cannot converge after exhausting the allocated trials and report these results with a (*) marker, signaling the challenge in tuning parameters for such algorithms. More details are available in Appendix. 7.4.

**Evaluation.**  Real robot evaluations can have high variance due to reset conditions and hardware noise. For each agent, we collect 12 trajectories and report their mean and standard deviation of scores. To confirm the reproducibility of our results and robustness to seed sweeping, for agents that contributed to our conclusions (usually the best and the second-best agents) we report performance swept over three consecutive random seeds (122, 124 in addition to the fixed seed of 123) in Appendix 7.6. To verify the statistical significance of our results when comparing performance between agents, we report the p-value of paired difference tests in Appendix 7.7.

**A. In-domain Ablations.**  We note the distinction between "in-domain" training and "out-domain" training, where the former leverages only data that were collected for the test task and the later allow incorporating heterogeneous data from different tasks. We first train all agents using in-domain data (i.e., we train a `slide` agent by feeding only `slide` data) to test ORL algorithms' sensitivity to varying data representation and inspect: (1) whether it is worth including velocity information in the state space (**Vel** versus **NoVel**); for simulator experiments, it is almost always a gain to include velocity, but velocity sensors on real robots are notoriously noisy; (2) whether to use the policy output joint position (**Abs**) vs the change in joint position (**Delta**) as action. Most BC literature uses the former, whereas RL prefers the later. [2] We use the outcome of the ablations and the best-performing setting for each algorithm to study generalization and transfer in the following three scenarios:

**B. Generalization: Lacking data support.**  The data collection may not cover the task space uniformly. For example, imagine that a robot trained to wipe clean a table but now cleans a bigger table. Empirically, a policy trained with behavior cloning would have trouble predicting actions for states when there is less data support. Can ORL algorithms, by learning a value function or model,

---

[2]Additionally, to verify that our dataset has reasonable optimality sufficient for training BC, we train BC separately with Top-K% of the trajectories to exclude the relatively "worse" trajectories. The results shown in Appendix. 7.5 verifies that BC has the best performance using the full dataset we collect.

| Task | Agent | Representations | | | |
|------|-------|---------|--------|------------|---------|
| | | **AbsNoVel** | **AbsVel** | **DeltaNoVel** | **DeltaVel** |
| **Reach** | BC | 0.863 ± 0.069 | 0.768 ± 0.118 | 0.912 ± 0.026 | **0.924 ± 0.048** |
| | Morel | 0.795 ± 0.086 | 0.584 ± 0.105 | 0.86 ± 0.069 | 0.917 ± 0.036 |
| | AWAC | 0.770 ± 0.105 | 0.713 ± 0.158 | 0.916 ± 0.030 | **0.925 ± 0.047** |
| | IQL | 0.843 ± 0.148 | 0.872 ± 0.104 | 0.904 ± 0.032 | 0.894 ± 0.066 |
| **Slide** | BC | 0.623 ± 0.172 | **0.681 ± 0.147** | 0.548 ± 0.200 | 0.551 ± 0.101 |
| | Morel | 0.356 ± 0.189 | 0.117 ± 0.235 | 0.532 ± 0.147 | 0.629 ± 0.160 |
| | AWAC | 0.548 ± 0.171 * | 0.591 ± 0.146 * | 0.569 ± 0.138 | 0.732 ± 0.113 |
| | IQL | 0.627 ± 0.144 | 0.589 ± 0.166 | 0.712 ± 0.137 | **0.767 ± 0.065** |
| **Lift** | BC | 0.759 ± 0.179 | **0.823 ± 0.177** | 0.721 ± 0.225 | 0.613 ± 0.142 |
| | Morel | 0.460 ± 0.189 | 0.149 ± 0.092 | 0.678 ± 0.186 | 0.652 ± 0.160 |
| | AWAC | 0.518 ± 0.083 * | <0 * | 0.863 ± 0.149 * | 0.821 ± 0.121 |
| | IQL | 0.682 ± 0.163 | <0 | 0.841 ± 0.144 | **0.880 ± 0.149** |
| **PnP** | BC | 0.632 ± 0.123 | **0.818 ± 0.185** | 0.564 ± 0.045 | 0.678 ± 0.195 |
| | Morel | <0 | <0 | **0.750 ± 0.197** | 0.748 ± 0.220 |
| | AWAC | 0.451 ± 0.159 * | <0 | 0.626 ± 0.234 * | 0.735 ± 0.175 * |
| | IQL | 0.469 ± 0.142 | <0 | 0.548 ± 0.160 | 0.601 ± 0.228 |

Table 2: Performance of all algorithms on varying representations. Each agent for each task is trained and evaluated on four settings: to include velocity in state or not (**Vel** versus **NoVel**); to use absolute or delta action space (**Abs** versus **Delta**). For each task, the best BC agent and the best ORL agent are highlighted and bolded. Agents that could not converge during training time are marked with (*). Some agents triggered violent crashes at test time and we report such performance as <0. Underline scores are swept over 3 seeds.

generalize to a task space that lacks data support? To this end, we create a new dataset from our slide task by dividing the task space to three regions: left, center, right. We remove any trajectory where the object was initially placed in the center region from the collected dataset. We train all agents and gather evaluation trajectories asking them to slide an object initially placed in the left, center and right regions.

**C. Generalization: Re-targeting data for dynamic tasks.** For the slide task, our collected demonstration has static goal positions. We test agents trained using such static data in a dynamic setting by updating the goal at a fixed frequency, and asking the agents to grasp and slide the lid following some predetermined curves. We collected the ideal trajectories via human demonstration. This task can be viewed as a simplified version of daily tasks, including drawing, wiping, and cleaning, which require possibly repeated actions and a much longer horizon than usual IL and ORL tasks. We select a variety of trajectories: circle, square, and the numbers 3, 5, 6, 8, which have different combinations of smooth curves and corners.

**D. Transfer: Reusing data from different tasks.** We investigate whether we can reuse heterogenuous data collected from previous tasks to train a policy for a new task. For example, would combining data from two canonical tasks (e.g., `slide+lift`) helps the agent perform better on either of these tasks? When aggregating data collected for multiple tasks, ORL algorithms can use the reward function for the test task to relabel the offline dataset. Evaluating ORL algorithms on such out-domain, transfer-learning settings is practical and relevant: instead of collecting random explorative data which demands careful setup of safety constraints on a real robot, we want to leverage offline datasets collected from different tasks to improve ORL performance. We train our algorithm with different combinations of canonical task demonstrations ("train-data") and evaluate each agent on each individual task.

| Start Position | BC | MOREL | AWAC | IQL |
|---|---|---|---|---|
| Left | 0.790 ± 0.056 | 0.571 ± 0.062 | 0.704 ± 0.119 | 0.704 ± 0.066 |
| Right | 0.774 ± 0.015 | 0.799 ± 0.033 | 0.707 ± 0.136 | 0.808 ± 0.015 |
| Center | 0.764 ± 0.013 | 0.793 ± 0.015 | 0.830 ± 0.026 | 0.811 ± 0.007 |
| Center, trained with full data | 0.791 ± 0.018 | 0.776 ± 0.022 | 0.813 ± 0.021 | 0.811 ± 0.050 |

Table 3: Training agents using a carved-out dataset to see how they perform when *generalizing* to a task region that lacks data support (the `Center` region, highlighted in Gray). For comparison, we also train all agents using full dataset and evaluate them on the `Center` region.

## 5 Results and Discussion

### 5.1 In-domain Tasks

**Which agent performs best for in-domain tasks?** Table 2 summarizes agents' performance for *in-domain* tasks, providing empirical insights into agents' sensitivity to varying representations. Interestingly, two of the ORL agent, IQL and AWAC achieved higher mean-scores than BC trained with abundant, in-domain real robot demonstrations on 2 out of 4 tasks. On the simplest task `reach`, the best version of all agents reached comparable performance. On the hardest task `PnP`, BC outperformed ORL agents. We would recommend considering IQL as a baseline even for imitation learning, when it is applicable to write a continuous reward function (an additional assumption compared to classic imitation learning).

**Sensitivity to representation** Empirically, BC demonstrated robustness to different state and action spaces, whereas ORL agents had varied performance. In 3 of 4 tasks considered, BCperformed the best when using absolution joint position as the action space and including velocity in the state space (**AbsVel**). On all tasks, ORL agents performed better using delta action space (**Delta**) rather than joint position. Intuitively, using the delta action space would be equivalent to restricting the policy to move in a unit ball centered around the current state. Such constraints could benefit RL policies that need exploration and sampling in action space more than it helped BC, which simply learns the mapping from states to actions. Additionally, we observed that our best agents all included velocity in their state space, though they were not guaranteed to have monotonic performance gain with velocity added.

### 5.2 Generalization and Transfer

**Generalization to regions that lack data support.** Table 3 trains agent using an carved-out dataset and compares the agents' performance on regions with more data support versus the region with less data support (`Center`, highlighted in Grey). For comparison, we also train all agents using full dataset and evaluate them on the `Center` region. On the region with abundant data support (`Left` and `Right`), we note that BC and IQL performed well and AWAC/MOREL got lower scores, aligning with our previous observation that BC/IQL performed better on in-domain tasks. Inspecting the performance on regions that have less data support (`Center`), however, we discovered that (1) AWAC and MOREL could match BC's performance on low-support region regardless of their initial disadvantage (they had poorer performance on the region with more support) and (2) ORL agents trained with carved-out dataset and evaluated on carved-out region performed no worse than them trained with full dataset, in contrast to BC agent, which performed worse after carving-out.

**Generalization to dynamic tasks.** Table 4 lists the ideal curves and the curves traced by each model. Each dot represents the location of the lid at a time step. BC had the worst performance among all models since it failed to complete tracing of the circle, square, and number 8 which requires a larger range of motion, and the BC agent seemed to get stuck during execution. Meanwhile, ORL methods largely succeeded tracing the entire curve following the time-varying goals, demonstrating stronger generalizing ability for this dynamic task.

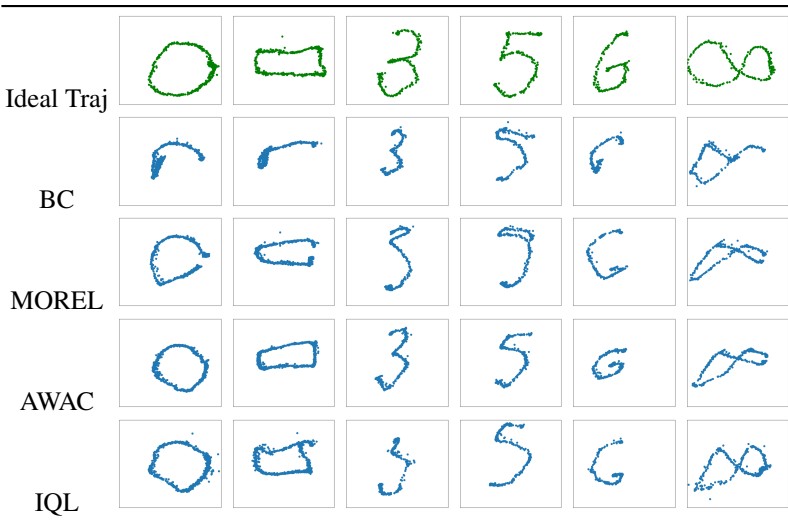

Table 4: Trajectory tracking. Green: the ideal demo trajectories, followed by each agent's tracking trajectories.

**Transfer learning by leveraging heterogeneous dataset**    Table 5 evaluates the performance of ORL algorithms when trained with different combinations of datasets from multiple tasks. We observe that:

1. The performance changes to ORL agents after leveraging offline data from different tasks can vary, due to the characteristics of the algorithm, the nature of the task, design of the reward function and the data distribution.

2. We observed all ORL agents *could* improve their own performance using some task/data combinations. Noticeably, MOREL achieved higher or comparable performance on all tasks after leveraging more offline data. For instance, its performance on the `lift` task progressively improved ($0.606 \rightarrow 0.726 \rightarrow 0.896$) with the inclusion of data from `slide` and `PnP` tasks. Intuitively, MOREL's dynamic model training process could benefit from any realistic data, regardless of whether the data was in-domain or out-of-domain.

3. Certain task-agnostic data could provide overlapping data support and enable effective transfer learning, allowing some ORL agents to surpass imitation learning and even the best in-domain agents. On `slide` and `lift`, all ORL algorithms managed to surpass BC. On PnP, AWAC achieved comparable performance as BCbut with a slightly higher mean using a combo of `slide` and `lift` data. With our extensive ablations, we observe that the final best agent for each task is either an ORL algorithm or a tie between ORL and BC.

4. However, ORL algorithms are not guaranteed to increase performance or guaranteed to surpass the best in-domain agents. The performance changes of ORL after leveraging out-domain data are likely to vary by agents, the task and dataset distribution. For instance, both AWAC and IQL agents have worse performance for `lift`, when using `slide+lift+PnP` than using only `slide+lift` data ($0.899 \rightarrow 0.728$, $0.863 \rightarrow 0.684$). Training IQL for `PnP` using `slide` or `slide+lift` data (without using `PnP` data), however, yielded even better results than using `PnP` data. Qualitatively we observe that IQL agents trained with `slide` data were better at grasping the object than the ones trained with `PnP` data at lifting the object, completing this first part of the task with more success while claiming distance-to-goal reward bonus.

With our extensive ablations, we observe that the final best agent for each task is either an ORL algorithm or a tie between ORL and BC.

**Random Seed Sweeping**    To improve the statistical significance of our results and to demonstrate reproducibility, we conduct additional random seed sweeping (i.e., train an agent with 3 consecutive random seeds). Results from additional agents (Appendix. 7.6) shows that the seed2seed variation of our experiments are low and provide more statistical significance to some of our observations: ∼60% of newly trained agents change score by less than 1%, ∼90% of agents change by less than 2%, and the maximum change was 6% from one agent (whose score change does not affect our conclusion).

| Agent | Train Data | Test Task | | |
|---|---|---|---|---|
| | | slide | lift | PnP |
| BC | in-domain | 0.681 ± 0.147 | 0.823 ± 0.177 | 0.818 ± 0.185 |
| | slide | 0.681 ± 0.147 | 0.582 ± 0.058 | 0.612 ± 0.083 |
| | slide+lift | 0.595 ± 0.127 | 0.580 ± 0.053 | 0.605 ± 0.120 |
| | slide+lift+PnP | 0.610 ± 0.137 | 0.609 ± 0.079 | 0.640 ± 0.144 |
| MOREL | in-domain | 0.629 ± 0.160 | 0.678 ± 0.186 | 0.750 ± 0.197 |
| | slide | 0.629 ± 0.160 | 0.606 ± 0.063 | 0.744 ± 0.174 |
| | slide+lift | 0.616 ± 0.146 | 0.726 ± 0.184 | 0.636 ± 0.173 |
| | slide+lift+PnP | 0.715 ± 0.134 | **0.896 ± 0.133** | 0.753 ± 0.181 |
| AWAC | in-domain | 0.732 ± 0.113 | 0.863 ± 0.149 * | 0.735 ± 0.175 * |
| | slide | 0.732 ± 0.113 | 0.638 ± 0.055 | 0.770 ± 0.111* |
| | slide+lift | 0.734 ± 0.110 * | 0.899 ± 0.149 | 0.813 ± 0.121 |
| | slide+lift+PnP | 0.644 ± 0.144 * | 0.728 ± 0.200 * | 0.758 ± 0.188 * |
| IQL | in-domain | **0.767 ± 0.065** | 0.880 ± 0.149 | 0.601 ± 0.228 |
| | slide | 0.767 ± 0.065 | 0.258 ± 0.033 | 0.810 ± 0.107 |
| | slide+lift | 0.704 ± 0.141 | 0.863 ± 0.166 | **0.842 ± 0.114** |
| | slide+lift+PnP | 0.643 ± 0.143 | 0.684 ± 0.158 | 0.833 ± 0.183 |

Table 5: Performance of agents trained with different combinations of offline data. The best in-domain agent, transfer learning agents that improves over their in-domain counterparts are colored. The **best agent for each task** is bold. Agents that could not converge during training time are marked with (*). Some agents triggered violent crashes at test time and we report such performance as <0. Underline scores are swept over 3 seeds.

**Comparison to previous works.**    Some of our *in-domain* conclusions are aligned with [40] and [39]: that behavior cloning demonstrates strong robustness to varying representations and tasks, serving as a competitive baseline in all four tasks tested. Even when BC is not the best, it has reasonable performance that is no worse than 85% of the best in-domain agents. Our findings also provide empirical verification to one of [34]'s observations that ORL could outperform BC for tasks where the initial state distributions change during deployment, a common condition for real robotic task, or when the environment has a few "critical" states, as seen in our manipulation tasks. In contrast to previous works, however, we highlight that (1) IQL can be a competitive baseline for settings that were traditionally favoring behavior cloning, as it turns out to be the best in-domain agent on 2 out of 4 tasks we tested, despite the lack of real robotic evaluation for IQL [18], (2) our extensive ablations on out-domain transfer learning are unique and allow us to verify several ORL algorithms' capability in generalizing to task region with less data-support (Table. 3) and to dynamic tasks (Figure. 4), (3) we observe that leveraging heterogeneous data has enabled all ORL algorithms to improve their own performance on at least one of the tasks, allowing some to even surpass the best in-domain agents, which suggest that ORL can be an interesting paradigm for real-world robotic learning.

## 6    Conclusion

In this work, we conducted an empirical study of representative ORL algorithms on real-world robotic learning tasks. The study encompassed three representative ORL algorithms (along with behavior cloning), four table-top manipulation tasks with a Franka-Panda robot arm, 3000+ train trajectories, 3500+ evaluation trajectories, and 270+ human labor hours. Through our extensive ablation studies, we find that (1) even for in-domain tasks with abundant amount of high-quality data, IQL can be a competitive baseline against the best behavior cloning policy, (2) for out-domain tasks, ORL algorithms were able to generalize to task regions with low data-support and to dynamic tasks, (3) the performance changes of ORL after leveraging heterogeneous data are likely to vary by agents, the design of the task, and the characteristics of the data, (4) certain heterogeneous task-agnostic data could provide overlapping data support and enable transfer learning, allowing ORL agents to improve their own performance and, in some cases, even surpass the best in-domain agents. Overall, (5) the best agent for each task is either an ORL algorithm or a tie between ORL and BC. Our rigorous empirical evaluations indicate that even in out-of-domain multi-task data regime, (more realistic in real world setting) offline RL is an effective paradigm to leverage out of domain data.

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

## 7 Appendix

### 7.1 Canonical Task Setup

We considered four canonical tasks: `reach`, `slide`, `lift` and PnP. To apply ORL, each task can be formulated as an MDP. The state contains the joint position of the robot, the gripper open position ($\mathcal{R} \sim [0, 0.08]$), (optionally) the velocity of the joints, (optionally) the tracked tag position and a goal position. To facilitate RL training, we came up with a continuous reward function for each task $r : state \rightarrow \mathcal{R}$, as shown in Table 6, considering the position of the gripper $x$, the position of tracked AprilTag $t$ (if exists), the position of goal $g$, the Euclidean distance function $dis$ between two 3D coordinates, a convenient function $height$ to denote the height of a given coordinates. While the reward for `reach` and `slide` are naturally smaller than 1, we explicitly cap the maximum reward for `lift` to be 1 since we don't encourage agents to lift up the lid arbitrarily high. We don't cap the PnP reward since we encourage the pick-n-place policy to be distinguished from the policy with a height bonus $height(t)$.

We used heuristic policies to collect the demonstration data, as described in Sec. 3. Our policies have a reasonable success rate *accomplishing* the task but is not designed to be optimal in solving the MDP. To evaluate and compare between agents, we instead report the maximum reward over the trajectory as a proxy of the task completion ("score"). We report our heuristic policies' accumulated reward average over trajectories and the score.

| Task | $r(s)$ | $\sum r(s)$ | Score |
|------|--------|-------------|-------|
| Reach | $1 - dis(g - x)$ | 173 | 0.99 |
| Slide | $1 - (2 * dis(g - t) + dis(t - x))$ | 223 | 0.93 |
| Lift | $min(1, 0.57 - dis(t - x) + height(t))$ | 167 | 1 |
| Pick-n-place | $1 - (dis(g - t) + 2 * dis(t - x)) * 0.9 + height(t)$ | 281 | 1.09 |

Table 6: Characteristics of task and collected data.

### 7.2 Dataset

In addition to the reward functions and statistics of our dataset, we also attach the score distribution on each task to demonstrate our dataset's overall quality. From Figure 3, we can see that the score distribution for each task skew heavily to the left, which means the datasets are suitable for imitation learning as well.

### 7.3 Open Source Code and Dataset

To remain anonymity we have only uploaded our collected dataset to here. Once accepted, we would share code and instructions on how to process and use our dataset.

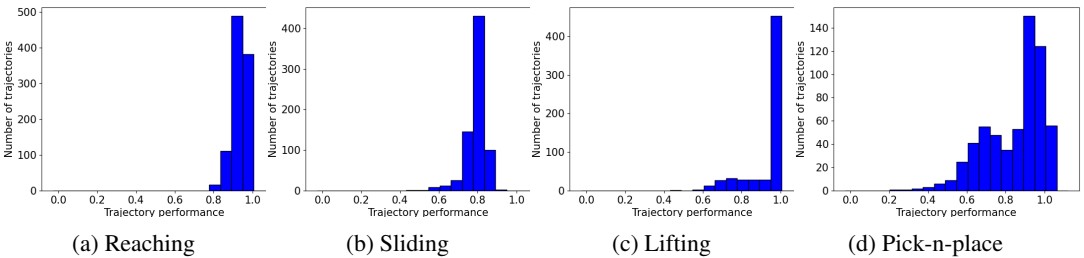

|        |        |        |        |        |
|--------|--------|--------|--------|--------|
| (a) Reaching | (b) Sliding | (c) Lifting | (d) Pick-n-place |

Figure 3: Score distribution for each task of our dataset.

## 7.4 Training Details

Our code base was built upon the author's implementation of MOREL [12] and the D3RLPY [56] library. We used the same fixed random seed for all our experiments, unless otherwise specified. For hyperparameter tuning, we always started by training using the default hyperparameters. If the training loss reported by the agent did not converge, we adjusted the learning rate and retrain, up to 5 agents, till we find a model that converge or have been trained for $5,000,000$ steps using batch size $2048$. For model whose training loss exploded (e.g., AWAC), we choose an checkpoint from earlier of the training when the loss were relatively stable for $100,000$ steps (frequently, this was an agent that finished about half a million to a million training steps). Surprisingly, when evaluated on real robot, models that reported convergence did not necessarily perform better than model that did not converge.

**Practicality of Training and Tuning**   BC was the cheapest to train ($\sim$ 3min) and easiest to converge (no additional tuning required). MOREL was the second shortest to train ($\sim$ 4 hours); most MOREL agents were able to converge, judged by the reward of trajectories generated by the learned dynamics model. AWAC agents took longer to train ($\sim$ 12 hours) and had the most trouble converging (8 of the 16 agents in the ablation table could not converge in allocated trials). IQL agents took the longest to train ($10 \sim 24$ hours) but had more success converging. Though loss convergence during training or a good reward estimated by the learned dynamics model or learned value function cannot indicate the agent's true performance, it is helpful for selecting an agent to test. Since some AWAC agents had trouble converging, we selected an earlier checkpoint before loss explosion and documented their performance, which, surprisingly, yielded higher reward than some agents that reported convergence. We leave it to future work to investigate this phenomenon.

## 7.5 Training Behavior Cloning with Top-K% Trajectories

To ensure that our dataset contains high quality trajectories that is sufficient to train behavior cloning, we launched new experiments training behavior cloning using only the Top-k % of the best trajectories. In Figure. 3, we plot the distribution of performance of our data for each task. For `reach`, `slide`, `lift`, 90% of trajectories complete the task with good scores ($> 0.75$). For (our most difficult task), 50% of our collected trajectories completed the task (scores $> 0.8$).

Thus we train BC for `reach`, `slide`, `lift` on Top-90% of data and train BC for `PnP` on Top-50%,70%,90% of data and observe that, BC in our experiments benefit from using the full dataset.

| Task | Top-k% | #Trajs | Threshold for Demo | Score | Original Score (BC with full data) |
|------|--------|--------|--------------------|-------|-------------------------------------|
| `reach` | 90 | 900 | 0.909 | $0.899 \pm 0.037$ | $0.924 \pm 0.048$ |
| `slide` | 90 | 657 | 0.774 | $0.659 \pm 0.152$ | $0.681 \pm 0.147$ |
| `lift` | 90 | 554 | 0.787 | $0.784 \pm 0.157$ | $0.823 \pm 0.177$ |
| `PnP` | 50 | 304 | 0.935 | $0.723 \pm 0.217$ | $0.818 \pm 0.185$ |
|  | 70 | 426 | 0.792 | $0.789 \pm 0.290$ | $0.818 \pm 0.185$ |
|  | 90 | 548 | 0.656 | $0.789 \pm 0.204$ | $0.818 \pm 0.185$ |

## 7.6 Sweeping of Random Seeds

We evaluated an addition of 28 agents for 340 trajectories for a total of 70 hours including training and testing to inspect how the scores for critical agents (i.e., the best agents for a category) would vary by random seeds. We now have 3 seeds for each of the following agents:

1. The Best Agents for each task in Table 2
2. The Second Best Agents for each task in Table 2
3. ORL agents with out-domain datasets in in Table 5

The original agents are trained with seed 123, we trained the additional agents with seed 122 and seed 124. Each seed is evaluated on 12 trajectories. The results are listed and we observe that ~60% of newly trained agents change score by less than 1%, ~90% of agents change by less than 2%, and the maximum change was 6% from one agent (whose score change does not affect our conclusion).

| Best Agents in Table 2 | Seed 122 | Seed 124 | Seed 123 (original seed) | Means w/ 3 seeds | Mean diff |
|---|---|---|---|---|---|
| AWAC, DeltaVel, `reach` | $0.920 \pm 0.031$ | $0.919 \pm 0.066$ | $0.935 \pm 0.032$ | $0.925 \pm 0.047$ | 0.01 (1.07%) |
| IQL, DeltaVel, `slide` | $0.781 \pm 0.038$ | $0.763 \pm 0.044$ | $0.757 \pm 0.095$ | $0.767 \pm 0.065$ | -0.01 (-1.32%) |
| IQL, DeltaVel, `lift` | $0.877 \pm 0.166$ | $0.878 \pm 0.158$ | $0.884 \pm 0.120$ | $0.880 \pm 0.149$ | 0.004 (0.45%) |
| BC, AbsVel, `PnP` | $0.819 \pm 0.199$ | $0.800 \pm 0.195$ | $0.836 \pm 0.157$ | $0.818 \pm 0.185$ | 0.018 (2.15%) |

| Second Best in Table 2 | Seed 122 | Seed 124 | Seed 123 (original seed) | Means w/ 3 seeds | Mean diff |
|---|---|---|---|---|---|
| MOREL, DeltaVel, `reach` | $0.919 \pm 0.034$ | $0.908 \pm 0.042$ | $0.925 \pm 0.028$ | $0.917 \pm 0.036$ | 0.008 (0.86%) |
| BC, DeltaVel, `reach` | $0.921 \pm 0.051$ | $0.917 \pm 0.055$ | $0.934 \pm 0.032$ | $0.924 \pm 0.048$ | 0.01 (1.07%) |
| BC, AbsVel, `slide` | $0.699 \pm 0.125$ | $0.698 \pm 0.120$ | $0.645 \pm 0.18$ | $0.681 \pm 0.147$ | -0.036 (-5.58%) |
| MOREL, DeltaVel, `slide` | $0.655 \pm 0.157$ | $0.602 \pm 0.180$ | $0.629 \pm 0.136$ | $0.629 \pm 0.160$ | 0 (0%) |
| AWAC, DeltaVel, `slide` | $0.757 \pm 0.068$ | $0.703 \pm 0.108$ | $0.739 \pm 0.144$ | $0.732 \pm 0.113$ | 0.007 (0.95%) |
| BC, AbsVel, `lift` | $0.821 \pm 0.192$ | $0.832 \pm 0.177$ | $0.818 \pm 0.161$ | $0.823 \pm 0.177$ | -0.005 (0.61%) |

| ORL in Table 5 | Seed 122 | Seed 124 | Seed 123 (original seed) | Means w/ 3 seeds | Mean diff |
|---|---|---|---|---|---|
| AWAC on `PnP` w/ slide+lift | (diverged) | $0.811 \pm 0.103$ | $0.815 \pm 0.134$ | $0.813 \pm 0.121$ | 0.002 (0.25%) |
| AWAC on `PnP` w/ slide+lift+pnp | $0.759 \pm 0.180$ | $0.773 \pm 0.204$ | $0.742 \pm 0.175$ | $0.758 \pm 0.188$ | -0.016 (-2.16%) |
| IQL on `PnP` w/ slide+lift | $0.838 \pm 0.103$ | $0.847 \pm 0.117$ | $0.843 \pm 0.120$ | $0.842 \pm 0.114$ | 0.001 (0.12%) |
| IQL on `PnP` w/ slide+lift+pnp | $0.842 \pm 0.170$ | $0.826 \pm 0.211$ | $0.829 \pm 0.163$ | $0.833 \pm 0.183$ | -0.004 (-0.48%) |
| MOREL on `lift` w/ slide+lift+pnp | $0.879 \pm 0.124$ | $0.904 \pm 0.119$ | $0.906 \pm 0.151$ | $0.896 \pm 0.133$ | -0.01 (-1.1%) |

### 7.7 Statistical Significance of Conclusions

In this section we verify the statistical significance of the conclusions we drew from our empirical study. To evaluate every trained agent for every task, we collected 12 trajectories and calculated their scores. One one hand, the estimated standard deviations of such scores were large, making the comparison between agents challenging (i.e. comparing $0.818 \pm 0.161$ with $0.884 \pm 0.120$). On the other hand, the distribution of scores is unknown. We cannot exclude the possibility of the distribution being skewed, as the agent could perform better in a certain task region because of the nature of the task. Therefore, we conducted both the dependent t-test ($p$) and the Wilcoxon signed T-test ($p_w$) for paired samples to calculate the p-value to reject or accept this null hypothesis: the two models' have identical scores.

We will reject the hypothesis with a small p-value ($p$ or $p_w < 0.1$). Tasks and application-domains determine the confidence level requirements for any application. This often requires domain knowledge and might not transfer between different applications even for the same task. For openness and interpretability, we clearly outline our statistical tests and list our p-values, leaving it up to the readers to justify their statistical significance required for their applications. We found that:

1. On in-domain tasks, we initially observe that: on `reach`, BC and the best ORL agent (AWAC) achieved similar performance ($0.93 \sim 0.93, p = 0.953, p_w = 0.844$); on `slide`, IQL outperform BC ($0.76 > 0.64, p = 0.066, p_w = 0.110$); on `lift`, we observe that BC is identical to the best ORL ($0.82 \sim 0.88, p = 0.146, p_w = 0.110$); on `PnP`, we observed that BC outperformed the best ORL agent ($0.90 > 0.75, p = 0.012, p_w = 0.016$). After running the best and the second best agents with multiple seeds, we can confirm the statistical significance of IQL outperforming BC on `lift` and `slide` ($0.88 > 0.82, p = 0.084, p_w = 0.041, 0.77 > 0.68, p = 0.001, p_w = 0.001$). With such observation, we recommend IQL and BC as a strong baseline for in-domain tasks.

2. Testing agent's ability to generalize to task space lacking data support, we verify that MOREL and AWAC achieved comparable performance or better to BC for regions lacking data support (MOREL: $0.80 \sim 0.77, p = 0.235, p_w = 0.500$, AWAC: $0.82 > 0.77, p = 0.006, p_w = 0.250$). It's worth noting that MOREL was having an initial disadvantage of having poorer performance on regions that have more data support ($0.67 < 0.78, p = 0.050, p_w = 0.062$).

3. In terms of leveraging task-agnostic data, MOREL has benefited from inclusion of more data. On Slide, the model achieved significantly higher performance when using combined data from three tasks $0.64 \sim 0.72, p = 0.113, p_w = 0.027$). On Lift, the model achieved significantly higher performance when using combined data from three tasks ($0.65 \rightarrow 0.91, p = 0.000, p_w = 0.003$). AWAC and IQL agents, however, had less success achieved higher scores. The only significant improvement is AWAC on Lifting ($0.82 \rightarrow 0.90, p = 0.082, p_w = 0.059$). Otherwise, AWAC agents performed the same irregardless of training data ($p > 0.1$). IQL had mostly similar or worse performance leveraging more data (e.g. worse slide performance: $0.70 \rightarrow 0.64, p = 0.069, p_w = 0.077$).

