# OpenReview forum: "Real World Offline Reinforcement Learning with Realistic Data Source"
_NeurIPS.cc/2022/Workshop/Offline_RL — Offline RL Workshop NeurIPS 2022_

### Official Review · Reviewer_Zb9x · 2022-10-13
**An informative benchmarking of offline RL algorithms**

**Rating:** 7
**Confidence:** 3

**Review:**

This paper benchmarks several offline RL algorithms with data generated by real robots. The benchmarking itself is of high quality. The paper is well written and easy to follow. Although benchmarking offline algorithms is not new, benchmarking them with real data appears novel to me and the offline RL community definitely benefits from such an empirical study.

---

### Official Review · Reviewer_22cA · 2022-10-18

**Rating:** 7
**Confidence:** 4

**Review:**

This paper presents an empirical study of offline reinforcement learning algorithms (and behavioral cloning) on 4 real-world tabletop manipulation tasks (reach, slide, lift, and pick-and-place), by collecting large datasets using scripted policies and evaluating how different models perform and generalize. I recommend acceptance for this paper - the datasets and empirical findings are very interesting and would be valuable to the larger community. That being said, I do think that given the simplicity of the tasks, the results should be taken with a grain of salt. Furthermore, it would have been incredibly useful to compare with the same tasks in simulation, in order to see whether there are important changes between sim and real that warrant real robot evaluations to take place - that seems to be one of the central claims in this paper, but it hasn't really been shown. It would be great to validate all of the hard work and effort that went into the collection of this large real world dataset.